Genome-wide characterization of endogenous retroviruses in snub-nosed monkeys

Wang Xiao 1
Wang Boshi 2
Liu Zhijin 2
Garber Paul A. 3
Pan Huijuan phjjanine@bjfu.edu.cn 1
1 Beijing Forestry University, School of Nature Conservation , Beijing , China
2 Chinese Academy of Sciences, Institute of Zoology, CAS Key laboratory of Animal Ecology and Conservation Biology , Beijing , China
3 University of Illinois, Department of Anthropology and Program in Ecology and Evolutionary Biology , Urbana , IL , America
Wilke Claus
Electronic publication date: 2019 Mar 18
Publication date: 2019
Volume: 7
Electronic Location ID: e6602
Received 2018 Oct 25; Accepted 2019 Feb 11
Copyright: ©2019 Wang et al.
Copyright year: 2019
Copyright holder: Wang et al.
License: This is an open access article distributed under the terms of the Creative Commons Attribution License, which permits unrestricted use, distribution, reproduction and adaptation in any medium and for any purpose provided that it is properly attributed. For attribution, the original author(s), title, publication source (PeerJ) and either DOI or URL of the article must be cited.
License URL: https://creativecommons.org/licenses/by/4.0/

Keywords: Endogenous Retrovirus, Rhinopithecus roxellanae, Rhinopithecus bieti, Classification

Funding: National Natural Science Foundation of China 31372185 31772438 This project was supported by grants from the National Natural Science Foundation of China (31372185 and 31772438). The funders had no role in study design, data collection and analysis, decision to publish, or preparation of the manuscript.

==============================
Background

Endogenous retroviruses (ERV) are remnants of former exogenous retroviruses that had previously invaded the germ line of the host that can be vertically transmitted across generations. While the majority of ERVs lack infectious capacity due to the accumulation of deleterious mutations, some ERVs remain active and produce potentially infectious viral particles. ERV sequences have been reported in all mammals; however, the distribution and diversity of ERVs in several primate taxa remains unclear. The aim of this study was to identify and classify the ERV sequences in the genomes of the golden snub-nosed monkey (Rhinopithecus roxellana) and the black and white snub-nosed monkey (Rhinopithecus bieti), two endangered primate species that exploit high altitude (2,500–4,500 m) temperate forests in southern and central China.

Methods

We used a TBLASTN program to search the ERV sequences of golden snub-nosed monkey genome and the black and white snub-nosed monkey genome. We retrieved all complete accession sequences from the homology search and then used the program, RetroTector, to check and identify the ERV sequences.

Results

We identified 284 and 263 endogenous retrovirus sequences in R. roxellana and R. bieti respectively. The proportion of full-length sequences of all ERV was 30% in R. roxellana and 21% in R. bieti and they were described as class I and class II or gamma-retrovirus and beta-retrovirus genera. The truncation pattern distribution in the two species was virtually identical. By analyzing and comparing ERV orthologues among 6 primate species, we identified the co-evolution of ERVs with their host. We also examined ERV-like sequences and found 48 such genes in R. roxellana and 63 in R. bieti. Some of those genes are associated with diseases, suggesting that ERVs might have involved the abnormal expression of certain genes that have contributed to deleterious consequences for the host.

Conclusions

Our results indicate that ERV sequences are widely distributed in snub-nosed monkeys, and their phylogenetic history can mirror that of their hosts over long evolutionary time scales. In addition, ERV sequences appear to have an important influence on the evolution of host pathology.

Introduction

Endogenous retroviruses (ERVs) are remnants of ancient retrovirus infections that have entered a species germ line and can be transmitted vertically to that host’s offspring (Wu et al., 2016). These sequences, have been detected in all vertebrates genomes including humans, and they often occupy a intermediate fraction of the genome (Blikstad et al., 2008; Gifford & Tristem, 2003). For example, ERVs contribute to approximately 8% of the human genome and 10% of the mouse genome (Griffiths, 2001; Subramanian et al., 2011; Wu et al., 2016).

Following recent classification, the International Committee on the Taxonomy of Viruses (ICTV) distinguishes seven genera within the retrovirus family: i.e., alpha-retrovirus, beta-retrovirus, delta-retrovirus, gamma-retrovirus, epsilon-retrovirus, spuma-virus and lenti-virus (Gifford & Tristem, 2003). The Retroviridae viral family can be divided into three classes, with gamma-retrovirus and epsilon-retrovirus are assigned to class I, beta-retrovirus, alpha-retrovirus, delta-retrovirus and lenti-virus are assigned to class II, and those clustering with the spuma-virus are assigned to class III (Stocking & Kozak, 2008; Vargiu et al., 2016; Villesen et al., 2004). Generally, a complete ERV element can be represented as 5′LTR-PBS-Gag-Pro-Pol-Env-PPT-3′LTR (Tongyoo et al., 2017). Two LTRs contain regulatory sequences that can alter the expression, splicing , and polyadenylation of those host genes located near the ERV insertion site (Hayward & Katzourakis, 2015). The LTRs are generated by reverse transcription and are attached to a primer binding site (PBS) and a polypurine tract (PPT). Gag genes encode the structural components of the viral core that includes the capsid (CA), matrix (MA) and nucleocapsid (NC) protein; Pro genes encode the viral protease (PR); Pol genes encode the reverse transcriptase (RT) and integrase (IN) enzymes; and Env genes encode the surface (SU) and transmembrane (TM) protein (Grandi & Tramontano, 2018b; Stocking & Kozak, 2008).

ERVs represent genomic fossils of past retroviral infections and can inform us of the diversity and history of retroviruses that have infected a species lineage (Zhuo, Rho & Feschotte, 2013). At present, studies of ERVs in primates have principally focused on the family Hominidae (humans and apes) and the subfamily cercopithecinae (certain African and Asian monkeys including baboons, macaques, guenons). There is growing evidence that ERVs have played an important role in the evolution of many mammalian lineages by providing new functions and evolutionary stimuli (Deininger et al., 2003; Grandi & Tramontano, 2018b; Heidmann et al., 2009). In humans, some HERVs are considered to be involved in various diseases. For example, the HERV-W Env mRNA was selectively upregulated in brain tissue from individuals with multiple sclerosis (Brutting et al., 2017; Christensen, 2017; Tselis, 2011). In addition, a significantly higher HERV-K10, MSRV and ERV-FRD activity was detected in the brains of patients with schizophrenia (Brutting et al., 2017). Although several studies have investigated the pathogenic role of HERVs, the precise link between any HERV sequence (and its expressed products) and human diseases still remains unclear (Grandi & Tramontano, 2018a). In general, the majority of research has focused on the characterization of ERV sequences and how this relates to the distribution and evolutionary history of primates. For example, BaEV, has been found only in baboons, geladas, mangabeys and in one subspecies of green monkey (Chlorocebus), indicating a strong phylogenetic relationship (Van der Kuyl, Dekker & Goudsmit, 1995). Similarity, the expressed level of one family of ERV, ERV-W Env, is significantly higher than other ERV families in the rhesus macaque (Macaca mulatta). This suggests that in the rhesus macaque, retroviral envelope proteins may play a more crucial role than the other ERV families in the evolutionary history of this species (Eo et al., 2014).

Although nonhuman primates represent model organisms for studies on endogenous retroviral diseases, studies of ERVs in the subfamily Colobinae are uncommon and lag behind other primate radiations. This likely results from the fact that given specializations of their digestive tract (colobines are foregut fermenters and possess a sacculated and low acid stomach that contains a highly diverse microbial community), and their consumption of a diet high in fiber, colobines are difficult to keep in captivity (Guo et al., 2018). Here we examine the distribution and classification of ERVs and function of ERV in two species of Asian colobines of the genus Rhinopithecus.

The Colobinae represent a highly successful radiation of 10 genera and over 76 species of African and Asian monkeys (IUCN, 2016) (Brandon-Jones et al., 2004). The sacculated stomach of colobines enable them to exploit difficult to digest foods that are high in structural carbohydrates and toxins such as tree bark, mature leaves, lichen, and seeds (Ross, 1995; Zhou et al., 2014). Snub-nosed monkeys comprise a group of five extant species of Asian colobines that are found in China (four species), Myanmar (one species) and Vietnam (one species). All five species are among the world’s rarest and most endangered primates (IUCN, 2000). The Chinese species of snub-nosed monkeys are unique because they are distributed in high altitude mountainous at elevations of from 2,000 to 4,500 m above sea level. In the case of R. roxellana and R. bieti, during the winter, snow may cover the ground for several months and nighttime temperatures drop below zero degrees centigrade (Guo et al., 2018; Long et al., 1994). A comparison of these two closely related species offers a framework to examine genetic and evolutionary differentiation that characterizes this primate lineage.

Recent advancements in genome sequencing offers the strongest method to sample ERV diversity. Complete genome sequence data allow us to investigate the distribution and diversity of ERV sequences within specific genomes in precise detail, and compare these details among different genomes (Gifford & Tristem, 2003). Here, we used bioinformatics tools to identify ERV sequences in R. roxellana and R. bieti using RetroTector (Sperber et al., 2007). We also identified neighboring genes of ERV sequences and discuss the possible effects of ERVs on gene regulation and their potential contribution in understanding how the structure and function of these sequences impact primate biology and health.

Materials & Methods

Identification of endogenous retrovirus

We used the Gag protein and the Env protein sequences of the gibbon ape leukemia virus (GenBank number: NP_056791.2; NP_056789.1) as a query and the TBLASTN program (Altschul et al., 1997) to search the golden snub-nosed monkey genome (GCA_000769185.1) and the black and white snub-nosed monkey genome (GCA_001698545.1). We obtained an extensive set of endogenous retroviruses in the sequenced genome and found approximately 100 hits for each species (the TBLASTN search threshold was set as 47% identity and total score was between 1,710 to 253). Then we uploaded the matched sequences to RetroTector online (ROL) to examine their structure and conducts a detailed analysis (Sperber et al., 2007). RetroTector (ReTe) is a platform-independent Java program for identification and characterization of provirus sequences in the vertebrate genome. The ROL is a light version of ReTe. ROL implementation (http://retrotector.neuro.uu.se/pub/queue.php), allows GenBank accession number, file, and FASTA cut-and-paste admission of sequences (5 to 1000000 kilobases). ROL can identify full integration and estimate the open reading frame. Moreover, it can detect proviruses a priori and is not dependent on repetition, giving one the capacity to identify low-copy number retroviral sequences (Sperber et al., 2009; Vargiu et al., 2016). This allowed us to obtain a detailed analysis of retroviral sequences found in the submitted sequences, which we then viewed with the program, RetroTectorViewer.jar.

Multiple sequence alignment and phylogenetic analysis

For a complete and accurate analysis, we selected a total of 141 complete full-length ERV sequences that contained 5′LTR, PBS, Gag, Pro, Pol, Env, PPT and 3′LTR structure, including 84 GERV sequences and 55 BERV sequences (Files S1–S4). A multiple alignment was constructed from the DNA sequences of the GERV and BERV using BioEdit v7.0.5 (Hall, 1999). Maximum likelihood (ML) phylogenies were estimated using nucleotide sequence alignments with Standard RAxML v8.1.17 (Stamatakis, 2014) with 1,000 bootstrap replicates. All ERV sequences were classified to different genera based on sequences similarity to known ERV sequences for vertebrates (Table 1). Phylogenetic trees were analyzed and adjusted using MEGA7 and iTOL v4 (Letunic & Bork, 2016)(http://itol.embl.de/).

Table 1 List of sequences used for phylogenetic analysis in this study.

Virus Abbreviation	Retrovirus	GenBank accession no.	Length	
Alpha retrovirus	
ALV	Avian leukosis virus	KU375453	7,746 bp	
RSV	Rous sarcoma virus	AF052428	9,396 bp	
LPDV	Lymphoproliferative disease virus	U09568	7,143 bp	
Beta retrovirus	
RERV	Rabbit endogenous retrovirus	AF480925	6,300 bp	
SRV-1	Simian SRV-1 type D retrovirus	M11841	8,173 bp	
MMTV	Mouse mammary tumor virus	NC_001503	8,805 bp	
Spuma retrovirus	
HFV	Human foamy virus	Y07725	13,242 bp	
FeFV	Feline foamy virus	AJ223851	10,479 bp	
SFV	Macaque simian foamy virus	X54482	12,972 bp	
SMRV	Squirrel monkey virus	GU356394	11,684 bp	
EFV	Equine foamy virus	AF201902	12,035 bp	
Delta retrovirus	
BLV	Bovine leukemia virus	K02120	8,714 bp	
HTLV-1	Human T-lymphotropic virus 1	D13784	8,400 bp	
STLV-1	Siman T-lymphotropic virus 1	Z46900	9,025 bp	
Epsilon retrovirus	
WDSV	Walleye dermal sarcoma virus	L41838	12,708 bp	
WEHV1	Walleye epidermal hyperplasia virus 1	AF133051	12,999 bp	
WEHV2	Walleye epidermal hyperplasia virus 2	AF133052	13,125 bp	
Gamma retrovirus	
BaEV	Baboon endogenous virus	X05470	8,018 bp	
MuLV	Moloney murine leukemia virus	AF033811	8,332 bp	
MDEV	Mus dunni endogenous virus	AF053745	8,655 bp	
GALV	Gibbon ape leukemia virus	M26927	8,088 bp	
FeLV	Feline leukemia virus	M18247	8,440 bp	
PERV	Porcine endogenous retrovirus	AF038601	7,333 bp	
KoRV	Koala type C endogenous virus	AF151794	8,431 bp	
RMLV	Rauscher murine leukemia virus	U94692	8,282 bp	
RaLV	Rat leukemia virus	M77194	8,107 bp	
Lentivirus				
HIV-1	Human immunodeficiency virus 1	K03454	9,176 bp	
HIV-2	Human immunodeficiency virus 2	X05291	9,671 bp	
SIV	simian immunodeficiency virus	M33262	10,535 bp	
FIV	Feline immunodeficiency virus	M25381	9,474 bp	
EIAV	Equine infectious anemia virus	AF028232	8,229 bp	
MVV	Ovine lentivirus	M31646	9,256 bp	
BIV	Bovine immunodeficiency virus	M32690	8,482 bp	

Identification of similar ERV sequences and Neighboring genes

Flanking sequences on both sides of the GERV and BERV sequences were selected to identify ERV-like sequences in the genome. The BLAST program was used to measure the similarity of all flanking sequences of GERV and BERV. We detected 13 sequences that were significantly similar with 0 e-value score and query cover ≥85%. These 13 GERV and BERV sequences also had high similarities. To discover additional ERV sequences in the whole genome and to reveal viruses that may have infected primates, we used ERV sequences (RR146/RB237) as a query sequence to query the whole genome sequences in primates with that of available whole genome sequences. For each potential ERV sequences, the RetroTector program was used to check their structure. We found ERV-like sequences in Homo sapiens (PRJNA168, GCA_000001405.27), Macaca fascicularis (PRJNA215851, GCA_000364345.1), Macaca mulatta (PRJNA16397, GCA_000772875.3), Piliocolobus tephrosceles (PRJNA419387, GCA_002776525.1), Theropithecus gelada (PRJNA477372, GCA_003255815.1) and Pan troglodytes (PRJNA10627, GCA_002880755.3). We used the same procedure indicated above to align sequences and build a phylogenetic tree.

The region information of genes and the position information of ERV sequences were obtained to determine the ERV neighboring genes. The flanking sequences of ERV were used to detect the homology genes in the human genome. Finally, we compared the similarities between the ERV with the flanking sequences as well as the same position of human genes obtained in the previous step by BLAST in NCBI.

Results

Endogenous retroviruses identification

Consistent with the consensus nomenclature used for the human endogenous retrovirus (HERV) (Boeke & Stoye, 1997) and the chimpanzee endogenous retrovirus (CERV) (Polavarapu, Bowen & Mcdonald, 2006), here we refer to the golden snub-nosed monkey endogenous retrovirus by the acronym GERV and black and white snub-nosed monkey endogenous retrovirus by the acronym BERV. Using the procedure described above, we identified a total of 284 GERVs with an average length of 8229 base-pairs and 263 BERVs with an average length of 8068 base-pairs (Files S3 and S4). For a complete and accurate analysis, we selected full-length ERV sequences which contained four major genes (gag, pro, pol and env), two sites critical to replication (PBS and PPT) and flanking with 5′LTR and 3′LTR (Files S1 and S2). We found 85 full-length GERVs with an average length of 9,364 base-pairs and 56 full-length BERVs with an average length of 8,867 base-pairs. The proportion of full-length sequences of all endogenous retroviruses was 30% in the golden snub-nosed monkey and 21% in black and white snub-nosed monkey.

According to the 5′LTR-PBS-Gag-Pro-Pol-Env-PPT-3′LTR structure of the endogenous retrovirus, we detected different types of truncation patterns. The proportion of each truncation pattern is shown in Fig. 1. Nine classification types of truncation patterns were detected: complete element, 5′-truncated, PBS truncated, gag truncated, pro truncated, pol truncated, env truncated, PPT truncated and 3′-truncated. As indicated in Fig. 1, the truncation pattern distribution in the golden snub-nosed monkey is similar to that in the black and white snub-nosed monkey. While the majority of truncation patterns in the golden snub-nosed monkey were PPT truncations and 3′-truncations, truncations in black and white snub-nosed monkey were enriched with PPT and PBS truncations. Overall, the fewest truncation patterns in each of the two species were gag truncations and pol truncations.

Figure 1 ERV truncation patterns distribution in the golden snub-nosed monkey (R. roxellana) and the black and white snub-nosed monkey (R. bieti).

Percentage of eight ERV truncation patterns and full-length ERV sequences in R. roxellana and R. bieti.

Grouping the endogenous retroviruses into classes and genera

We used 33 full-length endogenous retroviruses sequences (Table 1) to construct a phylogenetic tree using BioEdit v7.0.5 and RAxML v8.1.17. The resulting 33 endogenous retroviruses were grouped into three classes and seven genera based on the bootstrap values generated in the phylogenetic tree (Fig. 2). Class I contains elements related to the gamma-retroviruses and epsilon-retroviruses. Class II elements are related to alpha-retroviruses, beta-retroviruses, delta-retroviruses and lentiviruses. Class III elements are related to spuma-viruses.

Figure 2 Phylogenetic relationship of different ERVs.

Phylogenetic tree was based on 33 full-length ERV sequences in seven genera.

To better understand the classification of GERVs and BERVs, we further performed a phylogenetic analysis based on 84 full-length GERVs and 33 reference endogenous retrovirus sequences of seven genera (Fig. 3). Of the 84 GERVs identified in our study, 68 GERVs groups were class I which were related to gamma-retroviruses. Only seven GERVs groups in class II were related to beta-retroviruses. Class I GERVs accounted for 81% and class II GERVs accounted for 8.3%. Class I GERVs are the most abundant endogenous retroviruses in the golden snub-nosed monkey while only a small number of class II endogenous retroviruses were present. We also conducted a phylogenetic analysis based on 55 full-length BERVs and 33 reference endogenous retroviruses sequences characteristic of the seven genera (Fig. 4). In the tree shown, 21 BERVs groups of class I belonged to gamma-retroviruses genera, and 33 BERVs groups of class II belonged to beta-retroviruses. Class I BERVs accounted for 38.2% and class II BERVs accounted for 60%.

Figure 3 Phylogenetic relationship of different ERVs in golden snub-nosed monkey.

Phylogenetic tree was based on the 84 GERV sequences and 33 reference ERV sequences in seven genera. Within the tree, GERV sequences are named with RR (RR stands for the acronym of Rhinopithecus roxellana). The major taxonomic names are shown to the right.

Figure 4 Phylogenetic relationship of different ERVs in black and white snub-nosed monkey.

Phylogenetic tree was based on the 55 BERV sequences and 33 reference ERV sequences in seven genera. Within the tree, BERV sequences are named with RB (RB stands for the acronym of Rhinopithecus bieti). The major taxonomic names are shown to the right.

Identification of similar ERV sequences and inferring virus evolutionary history

Golden snub-nosed monkeys and black and white snub-nosed monkeys were found to exhibit several common ERV sequences, i.e., 13 GERV and BERV sequences that were virtually identical, including their flanking sequences. In order to study the evolutionary history of ERVs in primates, we used RR146/RB237 as a query sequence to search all primate genomes in the NCBI database. We found 13 species which had an ERV similar to RR146/RB237. However, only 6 of these species were detected by the RetroTector program. To understand their evolutionary relationships, we established a phylogenetic tree among ERV orthologues from Homo sapiens, Macaca fascicularis, Macaca mulatta, Piliocolobus tephrosceles, Theropithecus gelada and Pan troglodytes. The resulting phylogeny revealed two major lineages, the first containing the Old-World monkeys (Cercopithecoidea), including Macaca fascicularis, Macaca mulatta, Piliocolobus tephrosceles, Theropithecus gelada, Rhinopithecus roxellana and Rhinopithecus bieti (Fig. 5). The second lineage included Homo sapiens and Pan troglodytes, which belong to the Hominoidea. Given that our sampling of ERV diversity was limited, these results do not fully reflect the evolutionary history of the primate endogenous retroviruses. Moreover, based on our results (Fig. 5), the evolutionary history of ERVs tends to broadly follow the evolutionary history of their hosts. These data also revealed that virus phylogenetic history can mirror that of their hosts over long evolutionary timescales.

Figure 5 Phylogenetic relationship of similar ERVs.

Phylogenetic tree was based on the similar ERV sequences found in eight species and showed the evolutionary relationship of the similar ERV sequences in different primate species.

Endogenous retroviruses neighboring gene profiles

We found 48 and 63 neighboring genes in the golden snub-nosed monkey and black and white snub-nosed monkey, respectively (Files S5 and S6). By comparing these ERV neighboring gene sequences with their homologues in humans, we found, for example, the PEX3 gene, ALDH1A1 gene and ARL11 gene contain the ERV sequence in the golden snub-nosed monkeys, but not humans. In contrast, the PNPT1 gene and the DOCK5 gene, which contain ERV sequences, are identical in these two species. In the black and white snub-nosed monkey, the GPC5 gene and the TLR5 gene contain ERV sequences while in humans, these genes do not. However, the CHEK2 gene and DOCK5 gene in humans and in black and white snub-nosed monkey both contain an ERV sequence (Fig. 6). We also found that the DOCK5 gene is present in the golden snub-nosed monkey, the black and white snub-nosed monkey and humans, and the alignment results are highly similar.

Figure 6 Alignment of human and black and white snub-nosed monkey GPC5 and DOCK5 partial genes.

Graph showed the results of the black and white snub-nosed monkey and human genes alignment. (A) The partial sequence of the GPC5 gene was aligned between the black and white snub-nosed monkey and human. Query sequence was the GPC5 gene of the black and white snub-nosed monkey containing the ERV sequence. Human GPC5 gene did not contain ERV sequence. (B) Query sequence was the DOCK5 gene of the black and white snub-nosed monkey containing the ERV sequence. Human DOCK5 gene also contained ERV sequence.

Discussion

As part of the ERV infection process, a viral particle can become part of the germline of its host and then vertically transmitted to host’s offspring and future generations. Some elements are able to amplify within the genome and increase their copy number, leading to a larger number of ERV fragments (Dewannieux & Heidmann, 2013). Here, we have identified 284 ERVs in the genome of the golden snub-nosed monkey and 263 ERVs in the genome of the black and white snub-nosed monkey. The proportion of ERV fragments of all endogenous retroviruses was 70% in R. roxellana and 79% in R. bieti. We assume that the accumulation of nonsense mutations, insertions, and deletions of internal coding regions and long terminal repeats resulted in the inactivity of ERV sequences over evolutionary time, and those ERV sequences gradually become incomplete and exhibit many truncation patterns in the genome. Based on phylogenetic relationships (Figs. 3 and 4), we found that GERV sequences and BERV sequences mostly grouped with class I and class II ERVs. There was no ERV sequence related to delta-retroviruses or lenti-retroviruses. In addition, there was no ERV sequences in class III. This may be due to the fact that delta-retroviruses, spuma-retroviruses and lenti-retroviruses are all complex retroviruses. They code for regulatory proteins with different functions, in addition to gag, pro, pol and env proteins (Geng, 2012). A further explanation for why these genera have not been previously identified is that the trans-acting regulatory gene products of these complex genera have precluded germline integration (Gifford & Tristem, 2003). Alpha-retroviruses have only been identified previously in avians of the genus Gallus (e.g., pheasants) (Gifford & Tristem, 2003). Secondly, the query sequence we used in Blast is the Gag protein and Env protein sequence of the Gibbon ape leukemia virus, which is a gamma-retrovirus. Gamma-retroviruses and beta-retroviruses both have a simple genome organization. Finally, gamma-retroviruses are the most commonly found of all known retroviruses while alpha-retroviruses, delta-retroviruses and lenti-retroviruses are less common (Vargiu et al., 2016). Therefore, the proportion of gamma-retroviruses and beta-retroviruses in our results is relatively high.

ERVs are members of the LTR elements, which represent the most complex elements of retroelements (Khodosevich, Lebedev & Sverdlov, 2002). The LTRs of ERVs contain many regulatory sequences, such as promoters, enhancers, polyadenylation signals and factor-binding sites. Some of these ERVs could have integrated into regulatory regions of the genome, and they may influence the expression of nearby genes, which have consequently contributed to host evolution or associated with diseases (Goodier, 2016; Khodosevich, Lebedev & Sverdlov, 2002). This means that genes near the ERV could be affected by the promoter in the LTR of ERVs. For instance, HERV immunosuppressive functions might contribute to cancer progression by reducing the immune recognition and attack of tumor cells (Grandi & Tramontano, 2018b). Thus, ERVs we have found in the golden snub-nosed monkey and black and white snub-nosed monkey genomes may have influenced the expression of nearby genes and therefore associated with increased susceptibility to diseases. However, additional studies are needed to evaluate the potential impact of ERV expression on host health and its role in regulating signaling pathways involved in the manifestation of particular diseases.

Conclusions

Our analysis of the completed published golden snub-nosed monkey and black and white snub-nosed monkey genomes has identified 284 GERV and 263 BERV sequences, most of which are from genera of gamma-retroviruses and beta-retroviruses. The proportion of full-length sequences of all endogenous retroviruses is 30% in the golden snub-nosed monkey and 21% black and white snub-nosed monkey. However, more data are needed to determine whether these structurally complete ERV sequences remain active. In addition, the relationship between these ERV neighboring genes and diseases is unclear. In summary, our results document a co-evolutionary process between ERVs and their primate hosts that has occurred over the past several million years. Further analysis should allow us to better understand ERVs and their role in host protection.

Supplemental Information

File S1 Detailed information of ERV sequences in R. roxellana

The region, length, accession number and structure information of all ERV sequences in the golden snub-nosed monkey.

Click here for additional data file.

File S2 detailed information of ERV sequences in R. bieti

The region, length, accession number and structure information of all ERV sequences in the black and white snub-nosed monkey.

Click here for additional data file.

File S3 ERV sequences in R. roxellana

All ERV sequences found in the golden snub-nosed monkey.

Click here for additional data file.

File S4 ERV sequences in R. bieti

All ERV sequences found in the black and white snub-nosed monkey.

Click here for additional data file.

File S5 The list of ERV neighboring genes in R. roxellana

All ERV neighboring genes found in the golden snub-nosed monkey, including the corresponding ERV numbers and accession numbers.

Click here for additional data file.

File S6 The list of ERV neighboring genes in R. bieti

All ERV neighboring genes found in the black and white snub-nosed monkey, including the corresponding ERV numbers and accession numbers.

Click here for additional data file.

PAG wishes to thank Chrissie McKenney, Sara Garber, and Jenni Garber for their support.

Additional Information and Declarations

Competing Interests

Author Contributions

DNA Deposition

Data Availability

The authors declare there are no competing interests.

Xiao Wang conceived and designed the experiments, performed the experiments, analyzed the data, contributed reagents/materials/analysis tools, prepared figures and/or tables, authored or reviewed drafts of the paper, approved the final draft.

Boshi Wang conceived and designed the experiments.

Zhijin Liu contributed reagents/materials/analysis tools.

Paul A. Garber and Huijuan Pan authored or reviewed drafts of the paper.

The following information was supplied regarding the deposition of DNA sequences:

The raw data is available in the Supplemental Files.

The following information was supplied regarding data availability:

The detailed information of endogenous retroviruses sequences provided in Files S1 and S2.

The endogenous retroviruses data provided in the Files S3 and S4.

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
