# Peer review of "Genome-wide characterization of endogenous retroviruses in snub-nosed monkeys"

_PeerJ, doi:10.7717/peerj.6602_

## Round 0.1 · original submission · Major Revisions

Reviewer 1 raises major concerns about biases in the experimental design, insufficient detail in the methods, and insufficient discussion of relevant prior literature. These will have to be addressed with substantial revisions before we can consider the manuscript for publication. In addition, both reviewers express concerns about the quality of the writing, and this will also have to be addressed satisfactorily.

Reviewer 1 ·

Basic reporting

1) The English language should be improved throughout the manuscript, with particular attention to typing errors, plural vs singular nouns (especially when using possessive forms), and in the use of past tense. In addition, some sentences are unclear and should be rephrased. Some examples are given below:
“The vast majority (of) ERVs lack(s) infectious capacity due to the fact that they may accumulate as (accumulated) nonsense mutations, insertions, and deletions“
“envelop(e)”
“However, ERV in colobinae has remained unclear and lack of research.”
“Moreover, based on the resulting (Figure 5), the evolutionary of ERVs tends to broadly follow the evolutionary history of their host”
“LTR is one of the main groups of retroelements. ERVs, which are members of the LTR elements, represent the most complex of retroelements”

2) At page 8 lines 51-53 the authors report the association of ERVs with various diseases, stating “Some ERVs, however, have retained infectious properties and can be associated with diseases such as diabetes, some neoplasms, chronic diseases of the nervous system and autoimmune diseases (Brodziak et al.2012)”. This sentence, in my opinion, includes various misleading concepts. In fact, i) it seems that an ERV should produce infectious particles to have an effect, while it is known that most of the ERVs exert their effects as individual proteins or, more often, as noncoding RNA and cis-regulating agent, while there are no known example of any infection-competent ERVs (at least in humans); ii) authors claim various associations between ERVs and diseases, but up to date - despite the efforts done – no causal association has been demonstrated for any HERV in any disease. In general, while many studies detected the general expression of a given HERV group in a given diseased condition, very few of them identified specific HERV loci involved and/or a defined model of pathogenesis, preventing any corroborated causal association. Hence, the Reviewer thinks that such current lack of definitive etiological link should also be presented in the introduction, and suggests some recent reviews on the topic: i) HERV Envelope Proteins: Physiological Role and Pathogenic Potential in Cancer and Autoimmunity. Grandi and Tramontano 2018; ii) Human endogenous retroviruses in the aetiology of MS. Christensen 2017; iii) The role of molecular mimicry and other factors in the association of Human Endogenous Retroviruses and autoimmunity. Trela et al. 2016”;

3) At page 8, lines 55-56 Authors state that: “There is growing evidence that ERVs have played an important role in the evolution of many mammalian lineages including nonhuman and humans by providing new functions and evolutionary stimuli” and the same concept is reported again at page 9 lines 83-84 “Growing evidence suggests that ERVs have significantly contributed to primate evolution, diseases and development”. This is surely true but, being growing evidences, I would suggest to cite also more recent references given that the ones indicated date back to 10-15 years ago. Here some suggestions: i) Human Endogenous Retroviruses Are Ancient Acquired Elements Still Shaping Innate Immune Responses. Grandi and Tramontano 2018; ii) Transposable elements are the primary source of novelty in primate gene regulation. Trizzino et al. 2017; iii) Co-option of endogenous viral sequences for host cell function. Feschotte et al. 2017; iv) Regulatory activities of transposable elements: from conflicts to benefits. Chuong et al. 2017. In addition, given that the impact of ERVs on primate evolution is indicated as a main input for the work done, I think that the Authors should discuss more in details the main contribution of ERV sequences to primate physiology and evolution, allowing the readers to have a comprehensive overview of the findings’ significance.

4) At page 8, lines 58-64, authors present the general classification and taxonomy of Retroviridae family. This section should be rephrased avoiding repetitions (e.g.: the presence of 7 genera is mentioned two times in a couple of text lines) and should include all relevant information (e.g.: class II retroviruses includes not only Betaretroviruses, but also lentiviruses, alpha- and deltaretroviruses). I also suggest to cite more recent references for HERV classification and nomenclature, such as i) Classification and characterization of human endogenous retroviruses; mosaic forms are common. Vargiu et al. 2016; and ii) On the classification and evolution of endogenous retrovirus: human endogenous retroviruses may not be ‘human’ after all. Escalera-Zamudio and Greenwood 2016.

5) After a quick check, it seems that the article structure does not always conform to PeerJ standards. I suggest the Authors to check again the journal guidelines and to conform to their indication. Some examples: i) abstract should be organized in the following section indicated in bold: Background, Methods (not considered in the present abstract), and Results. ii) References sometimes lack the DOI number and the journal names are sometimes abbreviated and sometimes not. iii) all figures have no caption describing the results shown.

6) Some supplementary material was provided, but it was never referenced in the main text. Regarding the two excel files, I can suppose that one lists the sequences identified in R.roxellana and the other in R. bieti, but it is not clear which one is which because there are no titles or legends. In addition, the files contain some information about the ERV sequences found (such as length and identified genes) but does not provide their genomic coordinates, making impossible to localize them in the considered species genome without making individual blast searches.

Experimental design

1) In the Methods section, page 10 lines 110-116, Authors reported that to identify ERV sequences in the golden snub-nosed monkey genome (GCA_000769185.1) and the black and white snub-nosed monkey genome (GCA_001698545.1) they performed a TBLASTN using as queries the Gag protein and Env protein sequences of the gibbon ape leukemia virus (GenBank number: NP_056791.2; NP_056789.1). In my opinion, this constitute a relevant bias for the search outcome, because i) it excludes all ERV sequences devoid of Gag and Env coding portions, which could however have a physiological and/or pathological significance as well. In fact, it has been reported that also highly-defective ERVs and even solitary LTRs can exert biological functions; ii) it excludes all ERV lowly related to the gibbon ape leukemia virus in terms of structure, also considering that no cut-off threshold of identity and/or blast score is reported. In fact, authors only report that they downloaded 100 hits for each genome, that is the default number of results commonly displayed by the program.

2) In the result section, please use the proper nomenclature for viral genes (italics) and proteins (capitalized)

3) At page 13, lines 225 to 241, Authors found that 13 species showed an ERV sequence similar to a pair of proposed orthologs in the analysed monkey species (RR146/RB237). I think that this is quite expected, given the overall high similarity of primate ERVs from the same class, not necessarily indicating that they are orthologous loci. In addition, authors conclude that “results suggest that primates of a shared common ancestor may inherit the same ERV sequence from a common ancestor.” Also this concept is rather old and yet assessed, given that such a mechanism is at the bases of ERV spreading among primates. As an example, the ERV-W group has been exaustively characterized in the human genome (Contribution of type W human endogenous retrovirus to the human genome: characterization of HERV-W proviral insertions and processed pseudogenes. Grandi et al. 2016) and its individual members have been identified and compared one by one in 5 Catarrhini species, revealing that the great majority of them where shared among all these primates, being orthologous insertions (HERV-W group evolutionary history: characterization of the group in non-human primates and identification of highly related sequences in New World Monkeys. Grandi et al. 2018). Hence, I suggest the Authors to improve the conclusions taken from these results beside the general presence of shared loci.

4) At page 14 lines 244-255, Authors refer to “ERV-related genes around or within ERV sequences”. Related to what? The co-localization with ERV is not a relation and has not necessary a functional effect neither on the gene nor on the ERV sequence. Hence, the use of the expression “ERV-related genes” is misleading. Moreover, the fact that some monkey genes have the ERV insertions and the corresponding human genes do not does not provide any information, given that that specific ERV could have been deleted in humans, or have been integrated after the divergence between the two lineages. Hence, which conclusions can be taken from this result?

Validity of the findings

1) At page 15 lines 257-274, Authors should discuss the results relative to the ERV identification in the two monkey genomes considered. However, there is no mention of the results but only a repetition of the general ERV state of the art, including the accumulation of mutations and the general ERV structure. The latter is also imprecise, reporting the viral genes in the wrong order and omitting some proteins and functions (e.g. protease). I suggest the authors to rephrase the information and move most of it in the introduction, discussing here the ERV sequences identified during the study.

2) At page 16 lines 299-300, authors state that “The genes we found out indicate that the fusion of ERV with genes expressed by the host is ubiquitous”. This sentence contains various misleading concepts and should be removed or rephrased: i) what does it means “the fusion of ERVs with genes expressed by the host”? Are the two sequence fused together? If yes, how did authors established this? Did they checked for chimeric transcripts? ii) what does it means that the fusion is ubiquitous? Are all expressed genes co-localized with ERVs? Can authors provide the proportions of the intergenic and intragenic ERVs found?

3) At page 16 line 311 Authors state that “Some of the ERV-related genes we have identified appear to play a role in disease expression.” Did the Authors performed genome-wide association studies that are not reported in the present paper? Otherwise, how can they support this statement? Of course, the sole co-localization with human genes, even if involved in diseases, does not support in any way an involvement of the ERVs in any disease. Authors should state this clearly when claiming for uncertain associations.

4) Page 17 lines 335-345: as mentioned above (basic reporting section, point n° 2), to date, no definitive causal association has been demonstrated for any HERV in any disease, and neither specific HERV insertions nor a demonstrated mechanism of pathogenesis has been confirmed. Hence, the sentence “human endogenous retroviruses(HERV) activity has been strongly associated with ovarian, colon, gastric and pancreatic cancers and HERVs have also implicated in autoimmune diseases including rheumatoid arthritis(RA), multiple sclerosis(MS) and systemic lupus erythematosus(SLE)” is misleading and should be rephrased and completed referencing the relevant literature in the field.

Additional comments

The manuscript by Wang et al. analyze the presence of ERV sequences in two species of colobine primates, namely Rhinopithecus roxellana and Rhinopithecus bieti, through a Blast search and a subsequent structural and phylogenetic characterization. Overall, the manuscript has some potential, investigating two species for which ERVs have not been characterized yet. However, the methodological design is not sufficiently detailed and present some major biases, and both the results contents and discussion need extensive revision prior to publication. In addition, the presentation of the state of the art is often misleading, and the english language needs to be revised by a native speaker.

Reviewer 2 ·

Basic reporting

The authors performed comparative bioinformatic analysis of the endogenous retroviruses in golden snub-nosed monkey and the black and white snub-nosed monkey. The writing is clear except a few places, the background and discussion contains enough information, and the manuscript has standard sections. They also provided all the data and results within the main text and supplementary. The results also provide new insight into the evolutionary of endogenous retroviruses within primates.

Experimental design

The whole method is standard, and also provide sufficient information to repliacate.

Validity of the findings

The data and findings are robust, and they also provided the sequences of the endogenous retroviruses as supplementary files.

Additional comments

The authors performed comparative bioinformatic analysis of the endogenous retroviruses (ERV) in golden snub-nosed monkey and the black and white snub-nosed monkey. They found more than 200 ERV sequences in each monkey, and further identified the ERV related genes, which may play an important in the expression and also may be associated with diseases. I enjoyed reading the entire manuscript, and the science of this manuscript is sound for me, thus I think this work can be published with a few minor changes which are merely my personal perspectives.

1. The language should be improved. For example, there are many tense errors, e.g. Line 108, "discuss" should be discussed; Line 118, "download" should be downloaded. The whole writing is clear, but the authors should pay great effect to correct such errors.

2. The last section of the results. They found 48 and 63 ERV related genes in each monkey, and also discussed some of the genes in the Discussion. However, I could not see the whole list of these genes, please provide them as supplementary files.

3. Lines 311-333 in the Discussion. They described some important ERV related genes. This is not a problem, but I would like to see this part more than just simipliy telling us how important of those genes.

---

## Round 0.2 · accepted · Accept

Thank you for carefully revising your manuscript.

# Reviewer 2 ·

Basic reporting

no comment.

Experimental design

no comment.

Validity of the findings

no comment.

Additional comments

The authors have done a good job revising, and the paper is much clearer with the details and organization. I would fully recommend accepting the manuscript.